# Adaptive Salp Swarm Algorithm for Optimization of Geotechnical Structures



**Mohammad Khajehzadeh** [1] **, Amin Iraji** [2,*] **, Ali Majdi** [3] **, Suraparb Keawsawasvong** [4]
**and Moncef L. Nehdi** [5,*]

1   Department of Civil Engineering, Anar Branch, Islamic Azad University, Anar 77419-43615, Iran;
    mohammad.khajehzadeh@gmail.com
2   Engineering Faculty of Khoy, Urmia University of Technology, Urmia 57166-93188, Iran
3   Department of Building and Construction Techniques, Al-Mustaqbal University College, Hillah 51001, Iraq;
    alimajdi@mustaqbal-college.edu.iq
4   Department of Civil Engineering, Thammasat School of Engineering, Thammasat University,
    Bangkok 10200, Thailand; ksurapar@engr.tu.ac.th
5   Department of Civil Engineering, McMaster University, Hamilton, ON L8S 4M6, Canada
*   Correspondence: a.iraji@uut.ac.ir (A.I.); nehdim@mcmaster.ca (M.L.N.);
    Tel.: +1-905-525-9140 (ext. 23824) (M.L.N.)

**Abstract:** Based on the salp swarm algorithm (SSA), this paper proposes an efficient metaheuristic algorithm for solving global optimization problems and optimizing two commonly encountered geotechnical engineering structures: reinforced concrete cantilever retaining walls and shallow spread foundations. Two new equations for the leader- and followers-position-updating procedures were introduced in the proposed adaptive salp swarm optimization (ASSA). This change improved the algorithm's exploration capabilities while preventing it from converging prematurely. Benchmark test functions were used to confirm the proposed algorithm's performance, and the results were compared to the SSA and other effective optimization algorithms. A Wilcoxon's rank sum test was performed to evaluate the pairwise statistical performances of the algorithms, and it indicated the significant superiority of the ASSA. The new algorithm can also be used to optimize low-cost retaining walls and foundations. In the analysis and design procedures, both geotechnical and structural limit states were used. Two case studies of retaining walls and spread foundations were solved using the proposed methodology. According to the simulation results, ASSA outperforms alternative models and demonstrates the ability to produce better optimal solutions.

**Keywords:** salp swarm optimizer; spread foundation; retaining structures; economic design

## 1. Introduction

The objective function in most engineering problems is non-convex and discontinuous, with a large number of design variables. As a result, traditional deterministic optimization techniques based on mathematical principles may struggle to find a global optimum solution due to local optima trapping. The use of powerful metaheuristic optimization algorithms for obtaining a global optimum to overcome this limitation is of interest, and metaheuristic algorithms have proven to be an excellent alternative for solving complex problems in recent decades [1–6].

The most common geo-structures in practical application are reinforced concrete retaining walls and spread footings, which have received considerable attention in recent studies [7,8]. These structures are commonly used and typically involve a large volume of material. In the past, the initial anticipated dimensions of retaining structures were tested for stability and other building code requirements. If the dimensions were insufficient to meet the constraints, they were adjusted until all the requirements were met. During this time-consuming, iterative process, the cost of construction was not taken into account.

In the optimum design of these structures, the dimensions that provide the lowest cost and weight of construction while meeting all the design requirements are automatically determined.

Several metaheuristic algorithms for geotechnical engineering problems have recently been developed and are widely used. Despite the fact that metaheuristic methods can produce acceptable results, no algorithm outperforms another in solving all the optimization problems. Furthermore, the objective function in most geotechnical engineering optimization problems, such as shallow foundations, retaining structures, and pile optimization, is discontinuous and has a large number of design variables. As a result, several research projects have been launched in order to improve the performance and efficiency of the existing metaheuristics. Some of these are modified particle swarm optimizations [9,10], modified harmony search algorithms [11], modified gravitational search algorithms [12], modified sine cosine algorithms [13], improved salp swarm algorithms [14], modified ant colony optimizations [15], modified teaching–learning-based optimizations [16], improved tunicate swarm algorithms [17], and modified wild horse optimizations [18]. According to the effectiveness of the metaheuristics and their modified versions, these methods have been widely used to solve several geotechnical engineering problems, as presented in Table 1.

**Table 1.** Application of metaheuristic algorithms for geotechnical engineering problems.

| Author, Year | Reference | Optimization Method | Application |
|---|---|---|---|
| Goh, 2000 | [19] | Genetic algorithm | Locate the critical circular slip surface in slope stability analysis |
| Zolfaghari, Heath, and McCombie, 2005 | [20] | Genetic algorithm | Search for critical noncircular failure surface in slope stability analysis |
| Cheng et al., 2007 | [1] | Particle swarm optimization | Analyze two-dimensional slope stability |
| Cheng et al., 2008 | [11] | Improved harmony search algorithm | Analyze slope stability |
| Chan, Zhang, and Ng, 2009 | [21] | Hybrid genetic algorithms | Optimize pile groups |
| Kahatadeniya, Nanakorn, and Neaupane, 2009 | [22] | Ant colony optimization | Determine the critical failure surface of earth slope |
| Khajehzadeh et al., 2011 | [23] | Modified particle swarm optimization | Optimize design of spread footing and retaining wall |
| Camp and Akin, 2012 | [24] | Big bang–big crunch optimization | Optimize design of retaining wall |
| Camp and Assadollahi, 2013 | [25] | Hybrid big bang–big crunch algorithm | Optimize $CO_2$ and cost of reinforced concrete footings |
| Khajehzadeh et al., 2013 | [26] | Hybrid firefly algorithm | Multi-objective optimization of foundations |
| Kang, Li, and Ma, 2013 | [27] | Artificial bee colony algorithm | Locate the critical slip surface in slope stability analysis |
| Khajehzadeh, Taha, and Eslami, 2014 | [12] | Hybrid adaptive gravitational search algorithm | Multi-objective optimization of retaining walls |
| Kashani, Gandomi, and Mousavi, 2016 | [28] | Imperialistic competitive algorithm | Locate the critical slip surface of earth slope |
| Gordan et al., 2016 | [29] | Particle swarm optimization and neural network | Predict seismic slope stability |
| Gandomi and Kashani, 2017 | [7] | Accelerated particle swarm optimization, firefly algorithm, Levy-flight krill herd, whale optimization algorithm, ant lion optimizer, grey wolf optimizer, moth–flame optimization algorithm, and teaching–learning-based optimization algorithm | Minimize construction cost of shallow foundation |
| Aydogdu, 2017 | [30] | Biogeography-based optimization algorithm | Optimize cost of retaining wall |

**Table 1.** *Cont.*

| Author, Year | Reference | Optimization Method | Application |
|---|---|---|---|
| Gandomi et al., 2017 | [31] | Genetic algorithm, differential evolution, evolutionary strategy, and biogeography-based optimization | Analyze slope stability |
| Mahdiyar et al., 2017 | [32] | Monte Carlo simulation technique | Assess safety of slope |
| Gandomi, Kashani, and Zeighami, 2017 | [2] | Interior search algorithm | Optimize retaining wall |
| Chen et al., 2019 | [33] | Hybrid imperialist competitive algorithm and artificial neural network | Predict safety factor values of retaining walls |
| Koopialipoor et al., 2019 | [34] | Imperialist competitive algorithm, genetic algorithm, particle swarm optimization, and artificial bee colony combined with artificial neural network | Predict slope stability under static and dynamic conditions |
| Yang et al., 2019 | [35] | Neural network system | Design retaining wall structures based on smart and optimal systems |
| Xu et al., 2019 | [36] | Hybrid artificial neural network and ant colony optimization | Assess dynamic conditions of retaining wall structures |
| Himanshu and Burman, 2019 | [37] | Particle swarm optimization | Determine critical failure surface considering seepage and seismic loading |
| Kalemci et al., 2020 | [38] | Grey wolf optimization algorithm | Optimize retaining walls |
| Kaveh, Hamedani, and Bakhshpoori, 2020 | [39] | Eleven metaheuristic algorithms | Optimize design of cantilever retaining walls |
| Kashani et al., 2020 | [4] | Differential algorithm, evolution strategy, and biogeography-based optimization algorithm | Optimize design of shallow foundation |
| Sharma, Saha, and Lohar, 2021 | [40] | Hybrid butterfly and symbiosis organism search algorithm | Optimize retaining wall |
| Kaveh and Seddighian, 2021 | [41] | Black hole mechanics optimization, firefly algorithm, evolution strategy, and sine cosine algorithm | Optimize slope critical surfaces considering seepage and seismic effects |
| Temur, 2021 | [42] | Teaching–learning-based optimization | Optimize retaining wall |
| Li and Wu, 2021 | [43] | Improved salp swarm algorithm | Locate critical slip surface of slopes |
| Khajehzadeh, Keawsawasvong, et al., 2022 | [44] | Hybrid tunicate swarm algorithm and pattern search | Seismic analysis of earth slope |
| Arabali et al., 2022 | [45] | Adaptive tunicate swarm algorithm | Optimize construction cost and $CO_2$ emissions of shallow foundation |
| Khajehzadeh, Keawsawasvong, and Nehdi, 2022 | [46] | Artificial neural network combined with rat swarm optimization | Predict the ultimate bearing capacity of shallow foundations and their optimum design |
| Khajehzadeh, Kalhor, et al., 2022 | [47] | Adaptive sperm swarm optimization | Optimize design of retaining structures under seismic load |
| Kashani et al., 2022 | [48] | Multi-objective particle swarm optimization, multi-objective multiverse optimization and Pareto envelope-based selection algorithm | Multi-objective optimization of mechanically stabilized earth retaining wall |

A new meta-heuristic algorithm called the salp swarm algorithm (SSA) simulates salp fish swarming in deep waters [49]. Section 2 contains more information on the SSA's motivation and mathematical modelling. The SSA in its basic model can be extended or hybridized with another algorithm to produce better answers for future problems, similar to other metaheuristic approaches [14,43,50].

This paper presents an adaptive salp swarm algorithm (ASSA) for optimization by introducing new position-updating equations for leader and follower salps. This change significantly improves the algorithm's performance and convergence speed. A set of well-known standard benchmark functions from the literature is used to validate the

effectiveness of the proposed approach. Furthermore, numerical geotechnical structure optimization tests are used to investigate the proposed method's performance and efficiency.

## 2. Salp Swarm Algorithm

A salp is a type of marine animal in the Salpidae family. It has a cylindrical structure with apertures at the ends similar to those of a jellyfish, which move and eat by pumping water through internal feeding filters in their gelatinous bodies. The salp swarm algorithm (SSA), a population-based optimization technique, was developed by Mirjalili et al. [49]. The salp chain can be used to calculate the SSA's behavior while hunting for optimal feeding sources (i.e., the target of this swarm is a food position in the search space called FP). To mathematically model salp chains, they are sampled into two groups: followers and leaders. The salp at the head of the chain is known as the leader, while the others are known as followers. The swarm is led by the leader of these salps, and the followers follow in his footsteps. The chain begins with a leader, who is followed by the followers to guide their movements.

Similar to other swarm-based algorithms, the salp location is specified in a $n$-dimensional search space, where $n$ is the number of variables in a given problem. As a result, the positions of all the salps are recorded in a two-dimensional matrix known as $X$, as shown in Equation (1):

$$X_i = \begin{bmatrix} x_1^1 & x_2^1 & \cdots & x_d^1 \\ x_1^2 & x_2^2 & \cdots & x_d^2 \\ \vdots & \vdots & \cdots & \vdots \\ x_1^n & x_2^n & \cdots & x_d^n \end{bmatrix} \tag{1}$$

The fitness of each salp is then determined in order to define which salp has the best fitness. It is also supposed that the swarm's goal is a food position called $FP$ in the search area.

The following equation can be used by the leader salp to change positions:

$$x_i^1 = \begin{cases} FP_i + r_1((ub_i - lb_i)r_2 + lb_i) & r_3 \geq 0 \\ FP_i - r_1((ub_i - lb_i)r_2 + lb_i) & r_3 < 0 \end{cases} \tag{2}$$

where $x_i^1$ denotes the first salp's position in the $i$th dimension, and $FP_i$ denotes the food position in the $i$th dimension. The lower and upper bounds of the $i$th dimension are represented by $lb_i$ and $ub_i$, respectively, and the coefficient $r_1$ is calculated with Equation (3):

$$r_1 = 2e^{-\left(\frac{4t}{t_{max}}\right)^2} \tag{3}$$

In addition, the random numbers $r_2$ and $r_3$ are between 0 and 1. The maximum number of iterations is $t_{max}$, and the current iteration is $t$. It is worth noting that the $r_1$ coefficient is critical in a SSA because it balances exploration and exploitation throughout the search. The following equations are used to change the positions of the followers:

$$x_i^j = \frac{1}{2}\left(x_i^j + x_i^{j-1}\right) \tag{4}$$

where $j \geq 2$. In case some agents transfer outside of the search area, Equation (6) shows how to move salps back into the search area if they leave it:

$$x_i^j = \begin{cases} lb_i & if \ x_i^j \leq lb_i \\ ub_i & if \ x_i^j \geq ub_i \\ x_i^j & otherwise \end{cases} \tag{5}$$

The pseudocode of the SSA is shown in Algorithm 1.

---

**Algorithm 1.** Salp swarm algorithm

---

**Initialize** the salp population $x_i$ ($i$ = 1, 2, . . . , $n$) considering $lb_i$ and $ub_i$
**while** $t \leq t_{max}$
    **Calculate** the fitness of each search agent (salp)
    **Put** the best search agent as *FP* (Food position)
    **Update** $r_1$ by Equation (3)
      **for** each salp ($x_i$)
        **if** $i$ = 1
          **Update** the position of the leading salp by Equation (2)
        **else**
          **Update** the position of the follower salp by Equation (4)
        **end**
      **end**
    **Amend** the salps based on the upper and lower bounds of variables
   **Calculate** the fitness of each search agent *FP*
  **Update** the food position
  $t = t + 1$
 **end**
return the food position *FP* and its best fitness

---

## 3. Adaptive Salp Swarm Algorithm

Even though the SSA has the capability to generate acceptable results in comparison to other well-known techniques [49], the obtained results of the SSA are prone to becoming stuck in a local optimum, making it unsuitable for very complex problems with multiple local optima [43].

The leading salp adjusts its location in the SSA in response to the food situation (i.e., the position of the best salp in the whole population), as observed in Equation (2). The SSA algorithm updates the location of the leader salp around a single point at each incarnation pass, and other salps (followers) follow the leader. If the algorithm fails to recover because it lacks knowledge of the food position (FP), the algorithm fails. In other words, once an algorithm converges, it loses its ability to explore and then becomes inactive. As a result of this mechanism, the SSA algorithm becomes locked at local minimum points. In light of these circumstances, an adaptive version of the SSA (ASSA) is proposed to address the aforementioned flaw, while also increasing the algorithm's search capability and flexibility.

In the proposed ASSA, half of the population is considered as leaders, and the remaining salps are followers, which improves the algorithm's performance and exploring capabilities. The following equation is then used to update the position of the leader salps:

$$x_i^j = \begin{cases} x_i^j + r_1\left(FP_i - x_i^j\right) & r_3 \geq 0.5 \\ x_i^j - r_1\left(FP_i - x_i^j\right) & r_3 < 0.5 \end{cases} \tag{6}$$

The leaders adjust their positions in response to the state of the food source, as well as their previous position, as shown in Equation (6).

This process increases exploration while also allowing the SSA to conduct a more powerful global search across the entire search space. To improve the proposed ASSA's search efficiency, the followers update their positions according to the following equation:

$$x_i^j = rand^2\left(x_i^j + x_i^{j-1}\right) \tag{7}$$

In addition, in the suggested ASSA, at each iterative process, the worst salp with the highest objective function value is replaced with a completely random salp. The flowchart of the proposed ASSA is shown in Figure 1.

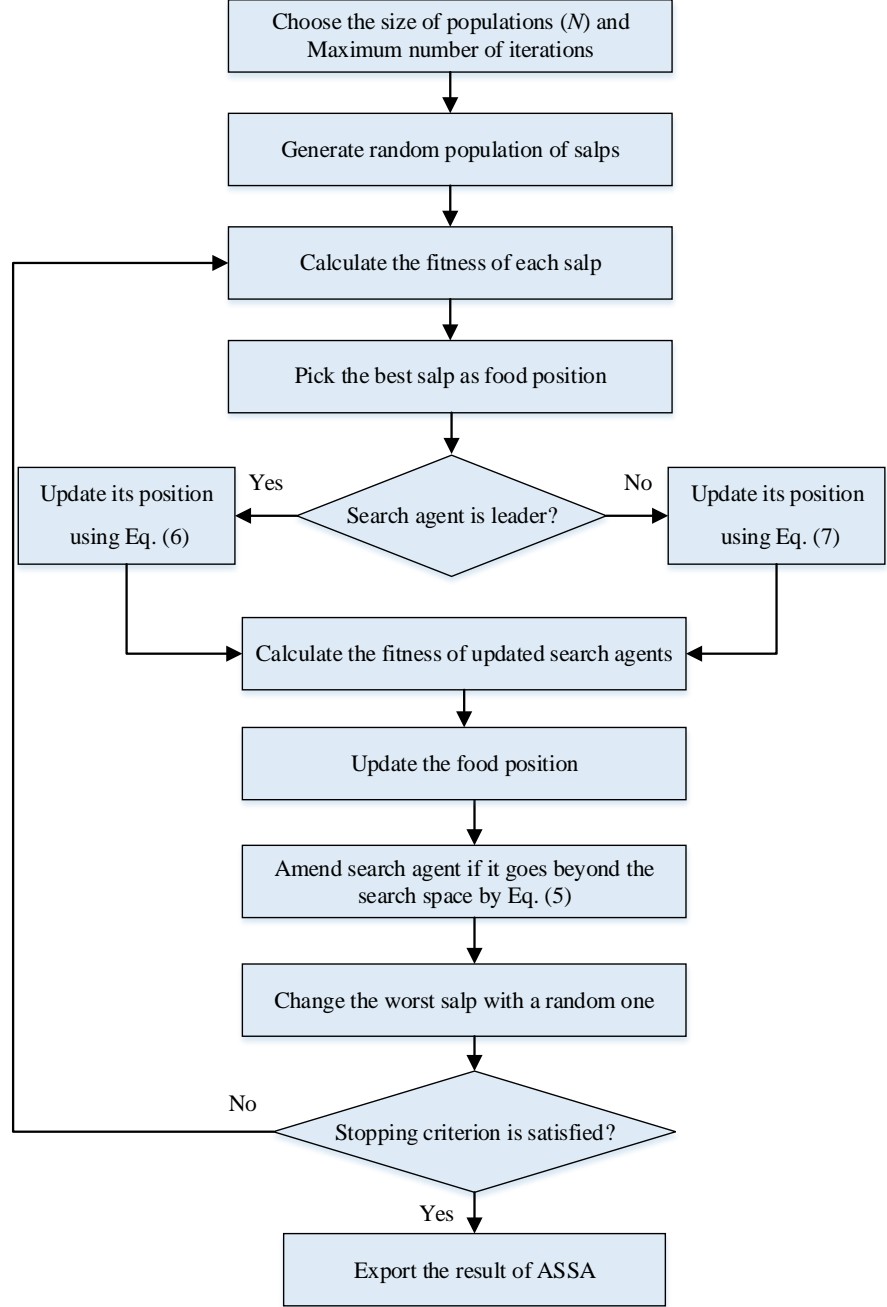

**Figure 1.** Flowchart of ASSA.

## 4. Model Verification

A set of numerical reference test functions was used in this section to compare and confirm the achievement and effectiveness of the proposed adaptive salp swarm algorithm (ASSA). In the empirical evidence literature, these functions have commonly been used to determine the performance of optimizers [51,52].

The mathematical models and characteristics of these test functions are shown in Tables 2 and 3. This standard set was divided into two categories: (1) unimodal functions with a single global best for testing the algorithm convergence pace and enslavement ability and (2) multimodal functions with multiple local minimums and a global ideal for testing an algorithm's local optima avoidance and exploratory capacity. MATLAB R2020b was used to create the suggested algorithms. All these functions should be minimized. Furthermore, all the functions had dimensions of 30. Three-dimensional drawings of these benchmark functions are illustrated in Figures 2 and 3.

The proposed ASSA was compared to the original SSA, as well as to some well-known optimization methods, such as the genetic algorithm developed by [53], the particle swarm optimization (PSO) proposed by [54], the firefly algorithm (FA) introduced by [55], the multiverse optimizer (MVO) developed by [56], and the tunicate swarm algorithm (TSA) introduced by [52]. For all methodologies, the sizes of the solutions (*N*) and the maximum number of iterations ($t_{max}$) were set to 30 and 1000, respectively, in order to make fair comparisons between them.

Because the results of a single run of a metaheuristic method are stochastic, they may be incorrect. As a result, statistical analysis should be performed in order to provide a fair comparison and evaluate an algorithm's efficacy. To address this issue, 30 runs for the mentioned methods were performed, with the results presented in Tables 4 and 5.

**Table 2.** Description of unimodal benchmark functions.

| Function | Range | $f_{min}$ | *n* (*Dim*) |
|:---:|:---:|:---:|:---:|
| $F_1(X) = \sum_{i=1}^{n} x_i^2$ | $[-100, 100]^n$ | 0 | 30 |
| $F_2(X) = \sum_{i=1}^{n} |x_i| + \prod_{i=1}^{n} |x_i|$ | $[-10, 10]^n$ | 0 | 30 |
| $F_3(X) = \sum_{i=1}^{n} \left( \sum_{j=1}^{i} x_j \right)^2$ | $[-100, 100]^n$ | 0 | 30 |
| $F_4(X) = \max_i \{ |x_i|, 1 \leq i \leq n \}$ | $[-100, 100]^n$ | 0 | 30 |
| $F_5(X) = \sum_{i=1}^{n-1} \left[ 100(x_{i+1} - x_i^2)^2 + (x_i - 1)^2 \right]$ | $[-30, 30]^n$ | 0 | 30 |
| $F_6(X) = \sum_{i=1}^{n} ([x_i + 0.5])^2$ | $[-100, 100]^n$ | 0 | 30 |
| $F_7(X) = \sum_{i=1}^{n} i x_i^4 + random[0, 1)$ | $[-1.28, 1.28]^n$ | 0 | 30 |

**Table 3.** Description of multimodal benchmark functions.

| Function | Range | $f_{min}$ | *n* (*Dim*) |
|:---:|:---:|:---:|:---:|
| $F_8(X) = \sum_{i=1}^{n} -x_i \sin\left( \sqrt{|x_i|} \right)$ | $[-500, 500]^n$ | $428.9829 \times n$ | 30 |
| $F_9(X) = \sum_{i=1}^{n} \left[ x_i^2 - 10\cos(2\pi x_i) + 10 \right]$ | $[-5.12, 5.12]^n$ | 0 | 30 |
| $F_{10}(X) = -20\exp\left( -0.2\sqrt{\frac{1}{n}\sum_{i=1}^{n} x_i^2} \right) - \exp\left( \frac{1}{n}\sum_{i=1}^{n}\cos(2\pi x_i) \right) + 20 + e$ | $[-32, 32]^n$ | 0 | 30 |
| $F_{11}(X) = \frac{1}{4000}\sum_{i=1}^{n} x_i^2 - \prod_{i=1}^{n}\cos\left( \frac{x_i}{\sqrt{i}} \right) + 1$ | $[-600, 600]^n$ | 0 | 30 |
| $F_{12}(X) = \frac{\pi}{n}\left\{ 10\sin(\pi y_1) + \sum_{i=1}^{n-1}(y_i - 1)^2 [1 + 10\sin^2(\pi y_{i+1})] + (y_n - 1)^2 \right\} + \sum_{i=1}^{n} u(x_i, 10, 100, 4)$ $y_i = 1 + \frac{x_i+4}{4}$ $u(x_i, a, k, m) = \begin{cases} k(x_i - a)^m & x_i > a \\ 0 & a < x_i < a \\ k(-x_i - a)^m & x_i < -a \end{cases}$ | $[-50, 50]^n$ | 0 | 30 |
| $F_{13}(X) = 0.1\{\sin^2(3\pi x_1) + \sum_{i=1}^{n}(x_i - 1)^2 [1 + \sin^2(3\pi x_i + 1)] + (x_n - 1)^2 [1 + \sin^2(2\pi x_n)] \} + \sum_{i=1}^{n} u(x_i, 5, 100, 4)$ | $[-50, 50]^n$ | 0 | 30 |

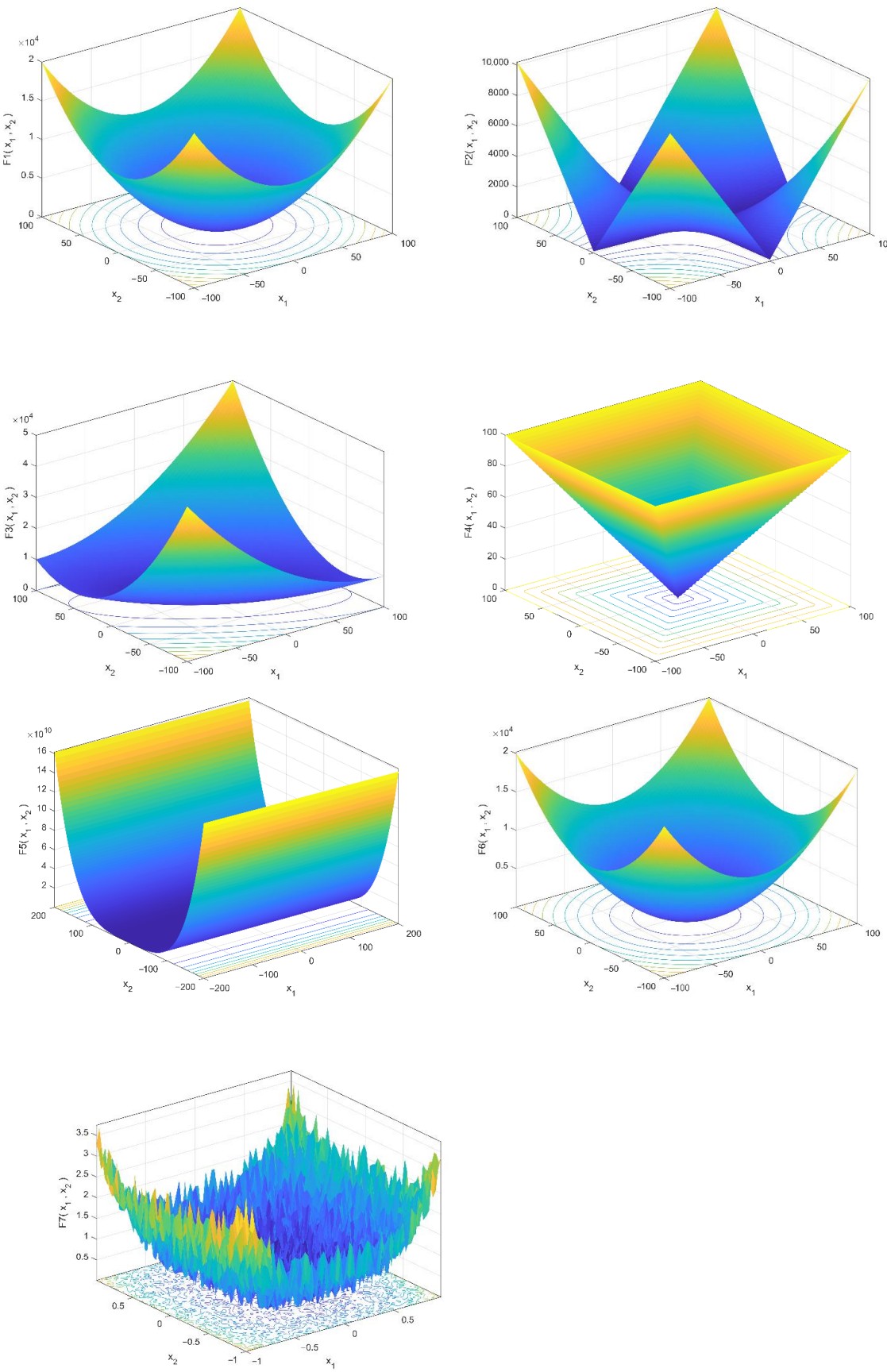

**Figure 2.** 3-D versions of unimodal benchmark functions.

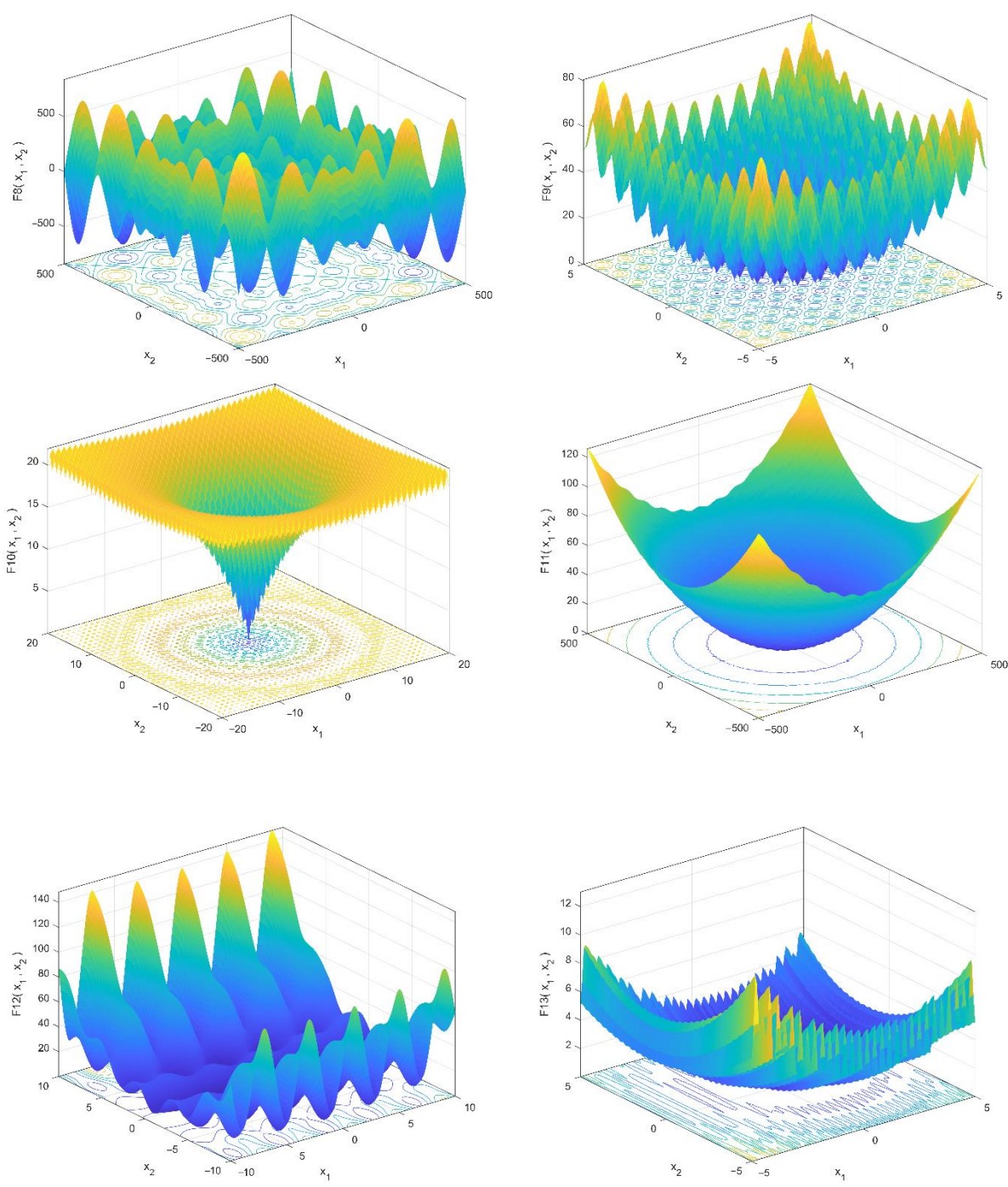

**Figure 3.** 3-D versions of multimodal benchmark functions.

**Table 4.** Comparison of different methods in solving unimodal test functions.

| F | Index | ASSA | SSA | GA | PSO | FA | MVO | TSA |
|---|---|---|---|---|---|---|---|---|
| $F_1$ | Mean | **$2.23 \times 10^{-227}$** | $3.29 \times 10^{-7}$ | $1.95 \times 10^{-12}$ | $4.98 \times 10^{-9}$ | $7.11 \times 10^{-3}$ | $2.81 \times 10^{-1}$ | $8.31 \times 10^{-56}$ |
| | Std. | **0.00** | $5.92 \times 10^{-7}$ | $2.01 \times 10^{-11}$ | $1.40 \times 10^{-8}$ | $3.21 \times 10^{-3}$ | $1.11 \times 10^{-1}$ | $1.02 \times 10^{-58}$ |
| $F_2$ | Mean | **$5.96 \times 10^{-105}$** | 1.911 | $6.53 \times 10^{-18}$ | $7.29 \times 10^{-4}$ | $4.34 \times 10^{-1}$ | $3.96 \times 10^{-1}$ | $8.36 \times 10^{-35}$ |
| | Std. | **$1.91 \times 10^{-104}$** | 1.614 | $5.10 \times 10^{-17}$ | $1.84 \times 10^{-3}$ | $1.84 \times 10^{-1}$ | $1.41 \times 10^{-1}$ | $9.86 \times 10^{-35}$ |
| $F_3$ | Mean | **$3.27 \times 10^{-180}$** | $1.50 \times 10^3$ | $7.70 \times 10^{-10}$ | 14.0 | $1.66 \times 10^3$ | 43.1 | $1.51 \times 10^{-14}$ |
| | Std. | **0.00** | $7.07 \times 10^2$ | $7.36 \times 10^{-9}$ | 7.13 | $6.72 \times 10^2$ | 8.97 | $6.55 \times 10^{-14}$ |
| $F_4$ | Mean | **$1.56 \times 10^{-104}$** | $2.44 \times 10^{-5}$ | 91.7 | $6.00 \times 10^{-1}$ | $1.11 \times 10^{-1}$ | $8.80 \times 10^{-1}$ | $1.95 \times 10^{-5}$ |
| | Std. | **$3.47 \times 10^{-105}$** | $1.89 \times 10^{-5}$ | 56.7 | $1.72 \times 10^{-1}$ | $4.75 \times 10^{-2}$ | $2.50 \times 10^{-1}$ | $4.49 \times 10^{-4}$ |
| $F_5$ | Mean | **$2.56 \times 10^{-1}$** | $1.36 \times 10^2$ | $5.57 \times 10^2$ | 49.3 | 79.7 | $1.18 \times 10^2$ | 28.4 |
| | Std. | **$4.78 \times 10^{-1}$** | $1.54 \times 10^2$ | 41.6 | 38.9 | 73.9 | $1.43 \times 10^2$ | $8.40 \times 10^{-1}$ |
| $F_6$ | Mean | **$3.76 \times 10^{-7}$** | $5.72 \times 10^{-7}$ | $3.15 \times 10^{-1}$ | $6.92 \times 10^{-2}$ | $6.94 \times 10^{-3}$ | $2.02 \times 10^{-2}$ | 3.67 |
| | Std. | **$1.23 \times 10^{-7}$** | $2.44 \times 10^{-7}$ | $9.98 \times 10^{-2}$ | $2.87 \times 10^{-2}$ | $3.61 \times 10^{-3}$ | $7.43 \times 10^{-3}$ | $3.35 \times 10^{-1}$ |
| $F_7$ | Mean | **$2.71 \times 10^{-6}$** | $8.82 \times 10^{-5}$ | $6.79 \times 10^{-4}$ | $8.94 \times 10^{-2}$ | $6.62 \times 10^{-2}$ | $5.24 \times 10^{-2}$ | $1.80 \times 10^{-3}$ |
| | Std. | **$2.33 \times 10^{-6}$** | $6.94 \times 10^{-5}$ | $3.29 \times 10^{-3}$ | $2.06 \times 10^{-2}$ | $4.23 \times 10^{-2}$ | $1.37 \times 10^{-2}$ | $4.62 \times 10^{-4}$ |

**Table 5.** Comparison of different methods in solving multimodal test functions.

| F | Index | ASSA | SSA | GA | PSO | FA | MVO | TSA |
|---|---|---|---|---|---|---|---|---|
| $F_8$ | Mean | **$-1.21 \times 10^4$** | $-7.46 \times 10^3$ | $-5.11 \times 10^3$ | $-6.01 \times 10^3$ | $-5.85 \times 10^3$ | $-6.92 \times 10^3$ | $-7.89 \times 10^3$ |
| | Std. | **$4.89 \times 10^2$** | $6.34 \times 10^2$ | $4.37 \times 10^2$ | $1.30 \times 10^3$ | $1.61 \times 10^3$ | $9.19 \times 10^2$ | 599.2 |
| $F_9$ | Mean | **0.00** | 55.45 | $1.23 \times 10^{-1}$ | 47.2 | 15.1 | $1.01 \times 10^2$ | 151.4 |
| | Std. | **0.00** | 18.27 | 41.1 | 10.3 | 12.5 | 18.9 | 35.87 |
| $F_{10}$ | Mean | **$8.88 \times 10^{-16}$** | 2.84 | $5.31 \times 10^{-11}$ | $3.86 \times 10^{-2}$ | $4.58 \times 10^{-2}$ | 1.15 | 2.409 |
| | Std. | **0.00** | $6.58 \times 10^{-1}$ | $1.11 \times 10^{-10}$ | $2.11 \times 10^{-1}$ | $1.20 \times 10^{-2}$ | $7.87 \times 10^{-1}$ | 1.392 |
| $F_{11}$ | Mean | **0.00** | $2.29 \times 10^{-1}$ | $3.31 \times 10^{-6}$ | $5.50 \times 10^{-3}$ | $4.23 \times 10^{-3}$ | $5.74 \times 10^{-1}$ | $7.7 \times 10^{-3}$ |
| | Std. | **0.00** | $1.29 \times 10^{-1}$ | $4.23 \times 10^{-5}$ | $7.39 \times 10^{-3}$ | $1.29 \times 10^{-3}$ | $1.12 \times 10^{-1}$ | $5.7 \times 10^{-3}$ |
| $F_{12}$ | Mean | **$2.31 \times 10^{-5}$** | 6.82 | $9.16 \times 10^{-8}$ | $1.05 \times 10^{-2}$ | $3.13 \times 10^{-4}$ | 1.27 | 6.373 |
| | Std. | **$2.46 \times 10^{-5}$** | 2.72 | $4.88 \times 10^{-7}$ | $2.06 \times 10^{-2}$ | $1.76 \times 10^{-4}$ | 1.02 | 3.458 |
| $F_{13}$ | Mean | **$1.44 \times 10^{-4}$** | 21.31 | $6.39 \times 10^{-2}$ | $4.03 \times 10^{-1}$ | $2.08 \times 10^{-3}$ | $6.60 \times 10^{-2}$ | 2.897 |
| | Std. | **$1.95 \times 10^{-4}$** | 16.99 | $4.49 \times 10^{-2}$ | $5.39 \times 10^{-1}$ | $9.62 \times 10^{-4}$ | $4.33 \times 10^{-2}$ | $6.43 \times 10^{-1}$ |

Tables 4 and 5 show that, for all the functions, the ASSA could provide better solutions in terms of mean value of the objective functions than the conventional SSA, as well as the other optimization techniques. The results also showed that the mean and standard deviation of the ASSA were significantly lower than those of the other strategies, indicating that the algorithm was stable. The ASSA outperformed both the standard method and alternative optimization approaches, according to the findings.

In order to obtain significant effectiveness between two or more algorithms, a nonparametric Wilcoxon's rank sum test is often used [57]. In this study, a pairwise comparison was performed using the best results from 30 runs of each algorithm. The Wilcoxon's rank sum test returned the *p*-value, the sum of the positive ranks (R+), and the sum of the negative ranks (R−). Table 6 shows the results of the Wilcoxon's rank sum test for all the benchmark functions. The *p*-value is the smallest level of significance for detecting differences. In this study, the level of significance was set at 0.05 ($\alpha = 0.05$). If the *p*-value was smaller than 0.05, it meant that the better result achieved by the best method in each pairwise comparison was statistically significant and was not obtained by chance. However, there was no significant difference between the two examined methods if the *p*-value was greater than 0.05. Such a result is indicated with "NA" in the "win" rows of Table 6. In addition, if the R+ was greater than the R−, the ASSA had a better performance than the alternative technique. Otherwise, the ASSA had a poor performance, and the other approach had a better performance [58].

**Table 6.** Results of Wilcoxon's rank sum test for benchmark functions.

| Fun. | Index | ASSA vs. SSA | ASSA vs. GA | ASSA vs. PSO | ASSA vs. FA | ASSA vs. MVO | ASSA vs. TSA |
|---|---|---|---|---|---|---|---|
| $F_1$ | $p$-val. | $2.0 \times 10^{-6}$ | $2.0 \times 10^{-6}$ | $2.0 \times 10^{-6}$ | $2.0 \times 10^{-6}$ | $2.0 \times 10^{-6}$ | $2.0 \times 10^{-6}$ |
| | R+ | 465 | 465 | 465 | 465 | 465 | 465 |
| | R− | 0.0 | 0.0 | 0.0 | 0.0 | 0.0 | 0.0 |
| | Win | ASSA | ASSA | ASSA | ASSA | ASSA | ASSA |
| $F_2$ | $p$-val. | $2.0 \times 10^{-6}$ | $2.0 \times 10^{-6}$ | $2.0 \times 10^{-6}$ | $2.0 \times 10^{-6}$ | $2.0 \times 10^{-6}$ | $2.0 \times 10^{-6}$ |
| | R+ | 465 | 465 | 465 | 465 | 465 | 465 |
| | R− | 0.0 | 0.0 | 0.0 | 0.0 | 0.0 | 0.0 |
| | Win | ASSA | ASSA | ASSA | ASSA | ASSA | ASSA |
| $F_3$ | $p$-val. | $2.0 \times 10^{-6}$ | $2.0 \times 10^{-6}$ | $2.0 \times 10^{-6}$ | $2.0 \times 10^{-6}$ | $2.0 \times 10^{-6}$ | $2.0 \times 10^{-6}$ |
| | R+ | 465 | 465 | 465 | 465 | 465 | 465 |
| | R− | 0.0 | 0.0 | 0.0 | 0.0 | 0.0 | 0.0 |
| | Win | ASSA | ASSA | ASSA | ASSA | ASSA | ASSA |
| $F_4$ | $p$-val. | $2.0 \times 10^{-6}$ | $2.0 \times 10^{-6}$ | $2.0 \times 10^{-6}$ | $2.0 \times 10^{-6}$ | $2.0 \times 10^{-6}$ | $2.0 \times 10^{-6}$ |
| | R+ | 465 | 465 | 465 | 465 | 465 | 465 |
| | R− | 0.0 | 0.0 | 0.0 | 0.0 | 0.0 | 0.0 |
| | Win | ASSA | ASSA | ASSA | ASSA | ASSA | ASSA |
| $F_5$ | $p$-val. | $2.0 \times 10^{-6}$ | $2.0 \times 10^{-6}$ | $2.0 \times 10^{-6}$ | $2.0 \times 10^{-6}$ | $2.0 \times 10^{-6}$ | $2.0 \times 10^{-6}$ |
| | R+ | 465 | 465 | 465 | 465 | 465 | 465 |
| | R− | 0.0 | 0.0 | 0.0 | 0.0 | 0.0 | 0.0 |
| | Win | ASSA | ASSA | ASSA | ASSA | ASSA | ASSA |
| $F_6$ | $p$-val. | $6.0 \times 10^{-6}$ | $2.0 \times 10^{-6}$ | $2.0 \times 10^{-6}$ | $2.0 \times 10^{-6}$ | $2.0 \times 10^{-6}$ | $2.0 \times 10^{-6}$ |
| | R+ | 453 | 465 | 465 | 465 | 465 | 465 |
| | R− | 12 | 0.0 | 0.0 | 0.0 | 0.0 | 0.0 |
| | Win | ASSA | ASSA | ASSA | ASSA | ASSA | ASSA |
| $F_7$ | $p$-val. | $6.0 \times 10^{-6}$ | $2.0 \times 10^{-6}$ | $2.0 \times 10^{-6}$ | $2.0 \times 10^{-6}$ | $2.0 \times 10^{-6}$ | $2.0 \times 10^{-6}$ |
| | R+ | 453 | 465 | 465 | 465 | 465 | 465 |
| | R− | 12 | 0.0 | 0.0 | 0.0 | 0.0 | 0.0 |
| | Win | ASSA | ASSA | ASSA | ASSA | ASSA | ASSA |
| $F_8$ | $p$-val. | $2.0 \times 10^{-6}$ | $2.0 \times 10^{-6}$ | $2.0 \times 10^{-6}$ | $2.0 \times 10^{-6}$ | $2.0 \times 10^{-6}$ | $2.0 \times 10^{-6}$ |
| | R+ | 465 | 465 | 465 | 465 | 465 | 465 |
| | R− | 0.0 | 0.0 | 0.0 | 0.0 | 0.0 | 0.0 |
| | Win | ASSA | ASSA | ASSA | ASSA | ASSA | ASSA |
| $F_9$ | $p$-val. | $2.0 \times 10^{-6}$ | $2.0 \times 10^{-6}$ | $2.0 \times 10^{-6}$ | $2.0 \times 10^{-6}$ | $2.0 \times 10^{-6}$ | $2.0 \times 10^{-6}$ |
| | R+ | 465 | 465 | 465 | 465 | 465 | 465 |
| | R− | 0.0 | 0.0 | 0.0 | 0.0 | 0.0 | 0.0 |
| | Win | ASSA | ASSA | ASSA | ASSA | ASSA | ASSA |
| $F_{10}$ | $p$-val. | $2.0 \times 10^{-6}$ | $2.0 \times 10^{-6}$ | $2.0 \times 10^{-6}$ | $2.0 \times 10^{-6}$ | $2.0 \times 10^{-6}$ | $2.0 \times 10^{-6}$ |
| | R+ | 465 | 465 | 465 | 465 | 465 | 465 |
| | R− | 0.0 | 0.0 | 0.0 | 0.0 | 0.0 | 0.0 |
| | Win | ASSA | ASSA | ASSA | ASSA | ASSA | ASSA |
| $F_{11}$ | $p$-val. | $2.0 \times 10^{-6}$ | $2.0 \times 10^{-6}$ | $2.0 \times 10^{-6}$ | $2.0 \times 10^{-6}$ | $2.0 \times 10^{-6}$ | $2.0 \times 10^{-6}$ |
| | R+ | 465 | 465 | 465 | 465 | 465 | 465 |
| | R− | 0.0 | 0.0 | 0.0 | 0.0 | 0.0 | 0.0 |
| | Win | ASSA | ASSA | ASSA | ASSA | ASSA | ASSA |
| $F_{12}$ | $p$-val. | $2.0 \times 10^{-6}$ | $2.0 \times 10^{-6}$ | $2.0 \times 10^{-6}$ | $2.0 \times 10^{-6}$ | $2.0 \times 10^{-6}$ | $2.0 \times 10^{-6}$ |
| | R+ | 465 | 0.0 | 465 | 465 | 465 | 465 |
| | R− | 0.0 | 465 | 0.0 | 0.0 | 0.0 | 0.0 |
| | Win | ASSA | GA | ASSA | ASSA | ASSA | ASSA |
| $F_{13}$ | $p$-val. | $2.0 \times 10^{-6}$ | $2.0 \times 10^{-6}$ | $2.0 \times 10^{-6}$ | $2.0 \times 10^{-6}$ | $2.0 \times 10^{-6}$ | $2.0 \times 10^{-6}$ |
| | R+ | 465 | 465 | 465 | 465 | 465 | 465 |
| | R− | 0.0 | 0.0 | 0.0 | 0.0 | 0.0 | 0.0 |
| | Win | ASSA | ASSA | ASSA | ASSA | ASSA | ASSA |
| | Superior /Inferior/NA | 13/0/0 | 12/1/0 | 13/0/0 | 13/0/0 | 13/0/0 | 13/0/0 |

According to the findings of the Wilcoxon's rank sum test in Table 6, the pairwise comparison of the ASSA and the SSA in the optimization of thirteen test functions demonstrated that the new approach outperformed the original method in all thirteen cases. Similarly,

in the other pairwise comparisons, the ASSA provided better results for the majority of the test suite. As a result of the nonparametric statistical analysis, the ASSA created much better answers and performed significantly better than the other techniques.

## 5. Foundation Optimization

A shallow spread foundation, as an essential geotechnical structure, must safely and reliably support the superstructure, guarantee stability against soil-bearing capacity failings and excessive settlement, and reduce concrete stresses. Aside from these design criteria, spread footings must meet a number of other criteria: they must have enough shear and moment capacities in both the long and short dimensions; the load-carrying capacity of the foundation must not be surpassed; and the reinforcing steel configuration must meet all building code criteria [59]. The foundation optimization problem requires determining the objective function, layout constraint, and design variables, which are discussed in the following subsections.

### 5.1. Objective Function

The total cost of the spread footing was the study's objective function, which can be expressed mathematically as follows:

$$f(X) = C_c V_c + C_e V_e + C_b V_b + C_f A_f + C_s W_s \tag{8}$$

In Equation (8), $C_c$, $C_e$, $C_b$, $C_f$, and $C_s$ are the unit costs of concrete, excavation, backfill, formwork, and reinforcement, respectively. The unit costs considered here are listed in Table 7 [60].

**Table 7.** Spread footing assembly unit cost [60].

| Item | Symbol | Unit | Unit Cost (USD) |
|---|---|---|---|
| Excavation | $C_e$ | m$^3$ | 25.16 |
| Formwork | $C_f$ | m$^2$ | 51.97 |
| Reinforcement | $C_s$ | kg | 2.16 |
| Concrete | $C_c$ | m$^3$ | 173.96 |
| Backfill | $C_b$ | m$^3$ | 3.97 |

### 5.2. Design Variables

Figure 4 depicts the design features for the given model. The design variables were divided into two categories: those that described geometric dimensions and those that described steel reinforcement. As shown in Figure 4, there were four spatial design variables that reflected the foundation dimensions: the foundation's length ($Y_1$), the width ($Y_2$), the thickness ($Y_3$), and the embedment's depth ($Y_4$). The steel reinforcement also had two design variables: the longitudinal reinforcement ($Y_5$) and the transverse reinforcement ($Y_6$).

### 5.3. Design Constraints

While optimizing and designing a reinforced concrete footing, both structural and geotechnical limit states should be considered. Two different geotechnical limit states are the bearing capacity of the surrounding geo-material and the permitted settlement of the footing. The shear capacity of the footing (one- and two-way shear), flexural capacity, and reinforcement limitation are all structural limit states. The structural limit states are investigated using ACI 318-11 specifications [59]. Service loads are commonly used to satisfy geotechnical limit states. Even so, factored loads can be used for structural limit states. Table 8 provides a list of both structural and geotechnical limit states.

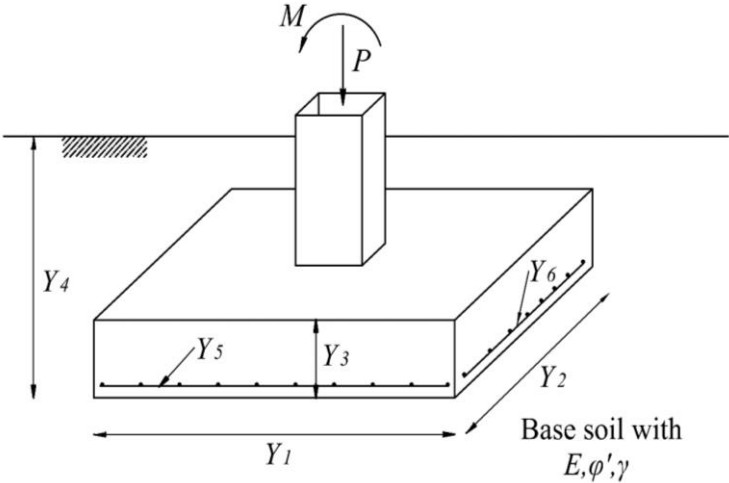

**Figure 4.** Design variables of the footing.

**Table 8.** Design constraints of spread footing.

| Failure Mode | Constraint |
|---|---|
| Bearing capacity | $q_{max} \leq \dfrac{q_{all}}{FS}$ |
| Settlement of foundation | $\delta \leq \delta_{all}$<br>$\delta = \dfrac{P(1-\mu^2)}{k_z E \sqrt{Y_1 Y_2}}$<br>$k_z = -0.0017 \left(\dfrac{Y_2}{Y_1}\right)^2 + 0.0597 \left(\dfrac{Y_2}{Y_1}\right) + 0.9843$ |
| Eccentricity | $e \leq Y_1/6$ |
| One-way (wide beam) shear | $V_u \leq \frac{1}{6} \varnothing_v \sqrt{f'_c} bd$ |
| Two-way (punching) shear | $V_u \leq$<br>$min\left\{\left(1+\frac{2}{\beta_c}\right)/6, \left(\frac{\alpha_s d}{b_0}+2\right)/12, \frac{1}{3}\right\} \varnothing_v \sqrt{f'_c} bd$ |
| Bending moment | $M_u \leq \varnothing_M A_s f_y \left(d - \frac{a}{2}\right)$ |
| Minimum and maximum reinforcements | $\rho_{min} bd \leq A_s \leq \rho_{max} bd$ |
| Limitation of depth of embedment | $0.5 \leq Y_4 \leq 2$ |

All the parameters presented in Table 8 are defined in Table 9.

**Table 9.** Definition of parameters of Table 7.

| Parameter | Definition |
|---|---|
| $q_{ult}$ | ultimate bearing capacity of the foundation soil |
| $q_{max}$ | maximum contact pressure at the interface between the bottom of a foundation and the underlying soil |
| $\delta_{all}$ | allowable settlement of foundation |
| $\delta$ | immediate settlement of foundation |
| $\phi_V$ | shear strength reduction factor equal to 0.75 |
| $f'_c$ | compression strength of concrete |
| $b_0$ | perimeter of critical section taken at $d/2$ from face of column |
| $b$ | width of the section |
| $\beta_c$ | ratio of long side to short side of column section |
| $\alpha_s$ | is equal to 40 for interior columns |
| $M_u$ | bending moment |
| $\phi_M$ | flexure strength reduction factor equal to 0.9 |
| $A_s$ | cross-sectional area of steel reinforcement |
| $f_y$ | yield strength of steel |
| $\rho_{min}$ | minimum reinforcement ratio |
| $\rho_{max}$ | maximum reinforcement ratio |

## 6. Retaining Structure Optimization

Reinforced concrete retaining walls are structures that are built to withstand lateral soil pressure as the land elevation changes. The retaining structure design process necessitates several considerations, such as structural dimensions, material characteristics, and needed reinforcement. Generally, the designer's experience plays a critical role in the cost-effective and safe design of these structures. However, the optimum design of retaining walls is independent of user experience, and the results satisfy both safety and economy.

### 6.1. Objective Functions

In the case of retaining structure optimization, the total construction cost of the retaining wall was considered as an objective function that incorporated the cost of materials, as well as labor and installation costs, that could be represented as follows:

$$f(X) = C_c V_c + C_e V_e + C_b V_b + C_f A_f + C_s W_s \tag{9}$$

In Equation (9), $C_c$, $C_e$, $C_b$, $C_f$, and $C_s$ are the unit costs of concrete, excavation, backfill, formwork, and reinforcement, respectively. Table 10 presents the unit construction of a retaining structure [61].

**Table 10.** Basic prices considered in the analysis.

| Item | Unit | Unit Cost (USD/m) |
|---|---|---|
| Excavation | $m^3$ | 11.41 |
| Foundation formwork | $m^2$ | 36.82 |
| Stem formwork | $m^2$ | 37.08 |
| Reinforcement | kg | 1.51 |
| Concrete in foundations | $m^3$ | 104.51 |
| Concrete in stem | $m^3$ | 118.05 |
| Backfill | $m^3$ | 38.10 |

### 6.2. Design Variables

Figure 5 depicts the retaining wall model's cross-section, design variables, and external load. As shown in this diagram, the dimensions of the retaining wall are represented by five geometric design variables: the heel width, represented by $X_1$; the top stem thickness, represented by $X_2$; the bottom stem thickness, represented by $X_3$; the toe width, represented by $X_4$; and the base slab thickness, represented by $X_5$. Three additional design features are included in the steel reinforcement of the various sections of the retaining wall. The vertical steel reinforcement in the stem is designated as $X_6$, the horizontal steel reinforcement in the toe is designated as $X_7$, and the horizontal steel reinforcement in the heel is designated as $X_8$. $B$ is the foundation's base width, $H$ is the wall's total height, and $H'$ is the stem's height.

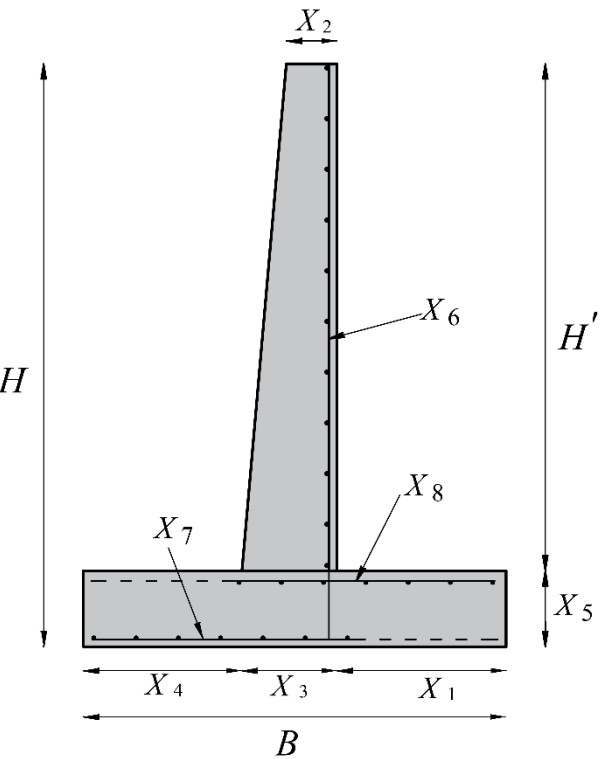

**Figure 5.** Design variables of the retaining structure.

### 6.3. Design Constraints

Figure 6 depicts the general forces acting on the retaining wall. Table 11 summarizes and presents the various design constraints that were taken into account when optimizing the concrete retaining wall.

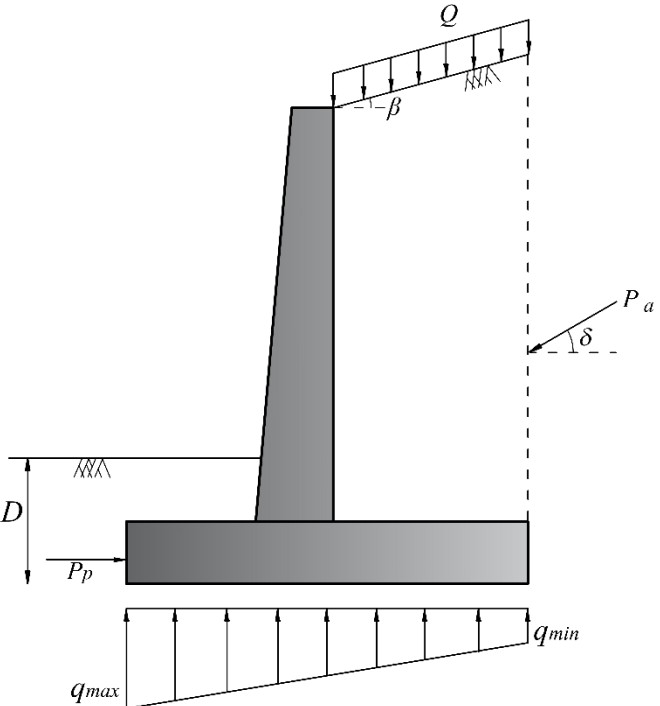

**Figure 6.** Forces acting on the retaining wall.

**Table 11.** Failure modes of retaining wall.

| Failure Mode | Constraints |
|---|---|
| Sliding stability | $FS_S \leq (\Sigma F_R / \Sigma F_d)$ |
| Overturning stability | $FS_O \leq (\Sigma M_R / \Sigma M_O)$ |
| Bearing capacity | $FS_b \leq (q_{ult} / q_{max})$ |
| Eccentricity failure | $e \leq (B/6)$ <br> $e = \frac{b}{2} - \frac{\Sigma M_R - \Sigma M_O}{\Sigma V}$ |
| Toe shear | $V_{ut} \leq V_{nt}$ |
| Toe moment | $M_{ut} \leq M_{nt}$ |
| Heel shear | $V_{uh} \leq V_{nh}$ |
| Heel moment | $M_{uh} \leq M_{nh}$ |
| Shear at bottom of stem | $V_{us} \leq V_{ns}$ |
| Moment at bottom of stem | $M_{us} \leq M_{ns}$ |
| Deflection at top of stem | $(1/150) \times H' \leq \delta_{max}$ |

All the parameters presented in Figure 6 and Table 11 are defined in Table 12.

**Table 12.** Definition of parameters of Figure 6 and Table 10.

| Parameter | Definition |
|---|---|
| $\beta$ | backfill slop angle |
| $D$ | depth of soil in front of the wall |
| $Q$ | distributed surcharge load |
| $P_a$ | active earth pressure |
| $P_p$ | passive earth pressure |
| $FS_S$ | required factor of safety against sliding |
| $FS_O$ | required factor of safety against overturning |
| $FS_b$ | required factor of safety against bearing capacity |
| $\Sigma F_R$ | sum of the horizontal resisting forces |
| $\Sigma F_d$ | sum of the horizontal driving forces |
| $\Sigma M_R$ | sum of the moments of forces that tends to resist overturning about the toe |
| $\Sigma M_O$ | sum of the moments of forces that tends to overturn the structure about the toe |
| $\Sigma V$ | sum of the vertical forces due to the weight of wall |
| $V_{ut}$ | ultimate shearing force of toe |
| $V_{uh}$ | ultimate shearing force of heel |
| $V_{us}$ | ultimate shearing force of stem |
| $V_n$ | nominal shear strength of concrete |
| $M_{ut}$ | ultimate bending moment of toe |
| $M_{uh}$ | ultimate bending moment of heel |
| $M_{us}$ | ultimate bending moment of toe stem |
| $M_n$ | nominal flexural strength of concrete |
| $\delta_{max}$ | maximum deflection at the top of the stem |

In addition to the constraints mentioned above, the design variables had practical lower and upper values [62]. Table 13 summarizes the lower and upper boundaries of the design variables.

**Table 13.** Upper bound and lower bound for design variables of retaining wall.

| Description | Lower Bound | Upper Bound |
|---|---|---|
| Width of footing | $B_{min} = 0.4\,H$ | $B_{max} = 0.7\,H$ |
| Thickness of base slab | $X_{5min} = H/12$ | $X_{5max} = H/10$ |
| Width of toe | $X_{4min} = 0.4\,H/3$ | $X_{4max} = 0.7\,H/3$ |
| Stem thickness at top | $X_{2min} = 20$ cm | - |
| Steel reinforcement ratio | $\rho_{min} = max\left\{\frac{1.4}{f_y}, 0.25\frac{\sqrt{f'_c}}{f_y}\right\}$ | $\rho_{max} = 0.85\beta_1\frac{f'_c}{f_y}\left(\frac{600}{600+f_y}\right)$ |

## 7. Design Examples

This section investigates numerical problems of geotechnical structures in order to evaluate the ASSA performance. To address the current inquiry, a MATLAB code was developed to computerize the design approach based on the ACI 318-11 specifications, as stated earlier [59].

In order to consider the constraints and transform a constrained optimization to an unconstrained problem, a penalty function method was used in this paper:

$$F(X) = f(X) + r\sum_{i=1}^{p} max\{0, g_i(X)\}^l \tag{10}$$

where $F(X)$ is the penalized objective function, $f(X)$ is the problem's original objective function presented in (8) and (9), and $g(X)$ is the problem's constraints presented in mboxcreftabref:applsci-1718195-t007,tabref:applsci-1718195-t010 for the spread footing and retaining wall, respectively. $r$ is a penalty factor considered equal to 1000, $l$ is the power of the penalty function considered equal to 2, and $p$ is the total number of constraints.

To demonstrate the efficacy of the proposed technique, the findings were compared to state-of-the-art algorithms such as particle swarm optimization (PSO) and the firefly algorithm (FA) in the following cases. The maximum number of iterations in any algorithm was assumed to be 1000. Because of the stochastic behavior of the metaheuristics in the following experiments, all the algorithms were run 30 times, and the best results of the analyses for the minimum cost obtained by each method are reported.

### 7.1. Spread Footing Optimization

The first two design examples were concerned with the best design for a dry sand inner surface spread footing. Table 14 lists the other input parameters for such case studies.

**Table 14.** Input parameters for design examples 1 and 2.

| Parameter | Unit | Value for Example 1 | Value for Example 2 |
|---|---|---|---|
| Effective friction angle of base soil | degree | 35 | 30 |
| Unit weight of base soil | kN/m$^3$ | 18.5 | 18.5 |
| Young's modulus | MPa | 50 | 35 |
| Poisson's ratio | — | 0.3 | 0.3 |
| Vertical dead load ($D$) | kN | 2000 | 4200 |
| Vertical live load ($L$) | kN | 1000 | 2100 |
| Moment ($M$) | kN-m | 0.0 | 850 |
| Concrete cover | cm | 7.0 | 7.0 |
| Yield strength of reinforcing steel | MPa | 400 | 400 |
| Compressive strength of concrete | MPa | 30 | 28 |
| Factor of safety for bearing capacity | — | 3.0 | 3.0 |
| Allowable settlement of footing | m | 0.04 | 0.04 |

The presented procedure solved the problem by combining all the previously mentioned algorithms. Tables 15 and 16 show the best results of the analyses for the lowest cost.

**Table 15.** Optimization results for design example 1.

| Design Variable | Unit | Optimum Values ASSA | Optimum Values SSA | Optimum Values FA | Optimum Values PSO |
|---|---|---|---|---|---|
| $(Y_1)$ | cm | 169.5 | 158.3 | 155.3 | 169.4 |
| $(Y_2)$ | cm | 218.8 | 248.5 | 253.1 | 219.2 |
| $(Y_3)$ | cm | 57.5 | 58.1 | 58.2 | 60 |
| $(Y_4)$ | cm | 200 | 158.2 | 200 | 200 |
| $(Y_5)$ | cm$^2$ | 39.58 | 48.2 | 49.65 | 37.75 |
| $(Y_6)$ | cm$^2$ | 25.13 | 21.74 | 20.94 | 23.91 |
| Objective function | USD | 1091 | 1098 | 1162 | 1108 |

**Table 16.** Optimization result for design example 2.

| Design Variable | Unit | Optimum Values ASSA | Optimum Values SSA | Optimum Values FA | Optimum Values PSO |
|---|---|---|---|---|---|
| $(Y_1)$ | cm | 153 | 153.1 | 159.3 | 153.2 |
| $(Y_2)$ | cm | 833.4 | 833.2 | 819.1 | 837.6 |
| $(Y_3)$ | cm | 80.6 | 80.6 | 82.4 | 80.8 |
| $(Y_4)$ | cm | 200 | 200 | 200 | 200 |
| $(Y_5)$ | cm$^2$ | 277.1 | 277.2 | 256.8 | 278.1 |
| $(Y_6)$ | cm$^2$ | 20.54 | 21.1 | 24.7 | 20.6 |
| Objective function | USD | 4512 | 4520 | 4650 | 4544 |

Tables 15 and 16 show that the optimization findings computed by the proposed ASSA were lower than those calculated by the conventional SSA and other approaches, indicating that the new method was effective. Table 15 shows that, contrary to popular belief that the best shape for a footing under vertical load is square, a rectangular footing provided a more cost-effective design.

### 7.2. Retaining Structure Optimization

The optimal design of two retaining walls with heights of 4 and 6 m was the subject of this section. Table 17 lists the other parameters that were required for this example.

**Table 17.** Input parameters for design examples 3 and 4.

| Parameter | Unit | Value for Example 3 | Value for Example 4 |
|---|---|---|---|
| Height of stem | m | 4.0 | 6 |
| Internal friction angle of retained soil | degree | 36 | 36 |
| Internal friction angle of base soil | degree | 0.0 | 34 |
| Unit weight of retained soil | kN/m$^3$ | 17.5 | 17.5 |
| Unit weight of base soil | kN/m$^3$ | 18.5 | 18.5 |
| Unit weight of concrete | kN/m$^3$ | 23.5 | 24 |
| Cohesion of base soil | kPa | 125 | 100 |
| Depth of soil in front of wall | m | 0.5 | 0.75 |
| Surcharge load | kPa | 20 | 30 |
| Backfill slop | degree | 10 | 15 |

**Table 17.** *Cont.*

| Parameter | Unit | Value for Example 3 | Value for Example 4 |
|---|---|---|---|
| Concrete cover | cm | 7.0 | 7.0 |
| Yield strength of reinforcing steel | MPa | 400 | 400 |
| Compressive strength of concrete | MPa | 21 | 28 |
| Shrinkage and temporary reinforcement percent | - | 0.002 | 0.002 |
| Factor of safety for overturning stability | - | 1.5 | 1.5 |
| Factor of safety against sliding | - | 1.5 | 1.5 |
| Factor of safety for bearing capacity | - | 3.0 | 3.0 |

Tables 18 and 19 show the results of the assessments for the examples with the lowest cost.

**Table 18.** Optimization results for design example 3.

| Design Variable | Unit | Optimum Values ASSA | Optimum Values SSA | Optimum Values FA | Optimum Values PSO |
|---|---|---|---|---|---|
| $(X_1)$ | m | 0.7233 | 0.6947 | 0.6948 | 0.6436 |
| $(X_2)$ | m | 0.2 | 0.2 | 0.2 | 0.25 |
| $(X_3)$ | m | 0.4674 | 0.5 | 0.5 | 0.55 |
| $(X_4)$ | m | 0.7778 | 0.7778 | 0.7778 | 0.7778 |
| $(X_5)$ | m | 0.2727 | 0.2727 | 0.2723 | 0.2727 |
| $(X_6)$ | cm$^2$/m | 6.67 | 6.66 | 6.66 | 6.66 |
| $(X_7)$ | cm$^2$/m | 6.75 | 6.75 | 6.75 | 6.75 |
| $(X_8)$ | cm$^2$/m | 6.75 | 6.75 | 6.75 | 6.75 |
| Objective function | USD/m | 822.73 | 827.02 | 860.42 | 848.17 |

**Table 19.** Optimization results for design example 4.

| Design Variable | Unit | Optimum Values ASSA | Optimum Values SSA | Optimum Values FA | Optimum Values PSO |
|---|---|---|---|---|---|
| $(X_1)$ | m | 1.423 | 1.391 | 1.459 | 1.444 |
| $(X_2)$ | m | 0.25 | 0.25 | 0.246 | 0.249 |
| $(X_3)$ | m | 0.531 | 0.532 | 0.466 | 0.517 |
| $(X_4)$ | m | 0.755 | 0.772 | 0.773 | 0.774 |
| $(X_5)$ | m | 0.331 | 0.374 | 0.352 | 0.339 |
| $(X_6)$ | cm$^2$/m | 25.38 | 25.64 | 32.21 | 27.52 |
| $(X_7)$ | cm$^2$/m | 6.78 | 6.75 | 6.75 | 7.02 |
| $(X_8)$ | cm$^2$/m | 7.94 | 7.47 | 7.57 | 8.39 |
| Objective function | USD/m | 1631.2 | 1643.1 | 1668.4 | 1653.9 |

Tables 18 and 19 show that, when compared to the traditional SSA and other methods, the ASSA may be able to provide a better solution by calculating lower values of the objective functions. It can be observed that the ASSA's best price was relatively lower than that of the SSA and significantly lower than that of the PSO and FA. However, additional experiments revealed that increasing the maximum number of iterations reduced the distinctions between the algorithm results. The fact that the best solution was found in the first iteration was due to the effective modification of the algorithm proposed in this study. The modified algorithm demonstrated a much-enhanced efficacy.

In order to determine the statistical significance of the comparative results between the considered algorithms in all the design examples, a nonparametric Wilcoxon's rank

sum test was performed between the results. In this regard, utilizing the results obtained from 30 runs of each method, a pairwise comparison was conducted. According to the results of the Wilcoxon's rank sum test in Table 20, the pairwise comparison between the ASSA and the SSA revealed that, in the optimization of four design examples, the new method had superior performances in three cases. In addition, for design example 3, both methods were statistically equivalent. Similarly, in the other pairwise comparisons, the ASSA provided better results. Therefore, the nonparametric statistical analysis proved that the ASSA generated significantly better solutions and, comparatively, had a superior performance over the other algorithms.

**Table 20.** Results of Wilcoxon's rank sum test for design examples.

| Example No. | Index | ASSA vs. SSA | ASSA vs. FA | ASSA vs. PSO |
|---|---|---|---|---|
| Ex. 1 | *p*-val. | $6.0 \times 10^{-6}$ | $1.73 \times 10^{-6}$ | $1.73 \times 10^{-6}$ |
| | R+ | 453 | 465 | 465 |
| | R− | 12 | 0.0 | 0.0 |
| | Win | ASSA | ASSA | ASSA |
| Ex. 2 | *p*-val. | 0.012 | $1.73 \times 10^{-6}$ | $1.73 \times 10^{-6}$ |
| | R+ | 354 | 465 | 465 |
| | R− | 111 | 0.0 | 0.0 |
| | Win | ASSA | ASSA | ASSA |
| Ex. 3 | *p*-val. | 0.106 | $1.73 \times 10^{-6}$ | $1.73 \times 10^{-6}$ |
| | R+ | 311 | 465 | 465 |
| | R− | 154 | 0.0 | 0.0 |
| | Win | NA | ASSA | ASSA |
| Ex. 4 | *p*-val. | $1.73 \times 10^{-6}$ | $1.73 \times 10^{-6}$ | $1.73 \times 10^{-6}$ |
| | R+ | 465 | 465 | 465 |
| | R− | 0.0 | 0.0 | 0.0 |
| | Win | ASSA | ASSA | ASSA |
| | Superior /Inferior/NA | 3/0/1 | 4/0/0 | 4/0/0 |

## 8. Conclusions

The primary objective of this study was to introduce an adaptive version of the salp swarm algorithm (ASSA). Two new equations for the leader- and follower-updating positions were introduced to improve the proposed ASSA's search and discovery abilities. In the standard SSA, the leading salp modifies its position based on a single point, which is the food location. However, due to a lack of knowledge about the real position of the food location, the algorithm may be locked at the local optimum. To overcome this weakness and to improve the exploration ability of the algorithm, in the proposed ASSA, half of the population was considered as leaders, which adjusted their positions not only based on the food location but also based on their previous positions. In addition, instead of the constant value considered in an SSA for follower-position-updating, in the ASSA, a random value was proposed. In addition, at each iteration of the optimization process, the ASSA replaced the worst salp, yielding the highest fitness value with a randomly generated salp. A statistical analysis was carried out in order to make an accurate assessment of the new algorithm's performance. The proposed method was shown to perform admirably in terms of accuracy, stability, and robustness when tested on some well-known unimodal and multimodal test functions. The paper's second goal was to automate a cost-effective design process for spread foundations and retaining walls. A computer program in Matlab was developed to reduce the cost of retaining structures and spread footings. On four case studies of these structures, the proposed method was compared to a classical SSA and some state-of-the-art metaheuristic algorithms. Given the final results, it was demonstrated that the ASSA outperformed the other techniques and should be able to provide better optimal

results. The new method concurrently satisfied geotechnical and structural limit states while simultaneously providing a cost-effective design.

**Author Contributions:** M.K.: methodology, software, and data curation. A.I.: investigation and writing—original draft preparation. A.M.: conceptualization and methodology. S.K.: resources and writing—original draft preparation. M.L.N.: supervision, project administration, validation, funding acquisition, and final draft preparation. All authors have read and agreed to the published version of the manuscript.

**Funding:** This research received no external funding.

**Institutional Review Board Statement:** Not applicable.

**Informed Consent Statement:** Not applicable.

**Data Availability Statement:** The data that support the findings of this study are available from the corresponding author upon request.

**Conflicts of Interest:** This research work abides by the highest standards of ethics, professionalism, and collegiality. All authors have no explicit or implicit conflict of interest of any kind related to this manuscript.

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
