# Peer review of "Adaptive Salp Swarm Algorithm for Optimization of Geotechnical Structures"

_applsci, doi:10.3390/app12136749_

Round 1

Reviewer 1 Report

The submitted paper can be interesting to readers, however in this form cannot be published.

Here are the reviewer remarks to be addressed in the paper:

  1. The title of the paper is too general. Instead of the "efficient metaheuristic algorithm" the specific "modified/adaptive salp swarp algorithm" should be used.
  2. The literature review on optimization in geotechnical applications as presented in the introduction is too short. Definitely more than seven references can be found in literature.
  3. The critique of the original SSA algorithm seems as if the SSA is only for local optimization, although in Ref. [16], there are many benchmarks which show the performance of the original SSA for multi-modal objective functions without being trapped into local minima. The claims in the submitted paper need evidence.
  4. Also the claim that the proposed SSA is adaptive may seem that the original SSA does not have adaptive mechanism, which is not the case. In fact the c1 (r1) parameter is used for exploration and exploitation adaptability.
  5. The r1 parameter (equation 3) should be placed in the text before the sentence describing parameters r2 and r3.
  6. Is the pseudo-code of Algorithm 1 developed by the submitting authors or copied/adapted from [16]?
  7. Again the claims in Section 3 (lines 140-142) are in contradiction in what is shown in Ref [16] where 13 benchmark examples does not pose premature local optimization.
  8. The figure 3 is not readable and the graphics quality is poor. It is not clear to what the X1, X3, X4 design variables refer?
  9. What are the bounds of all X design variables?
  10. The values in Table 10 are suspicious. The authors claim that the ASSA has the accuracy of e-227 while all other algorithms are significantly poorer. Based on the reviewer experience the numbers depend on the algorithm parameters, perhaps the ASSA parameters were optimally chosen while the parameters of other algorithms were not adjusted properly.
  11. Among the state-of-the-art optimization algorithms the classic genetic algorithm is missing which in many variants proves to perform well in the selected benchmarks.
  12. The value of 28.8 in Table 16 for X7 and FA algorithm seems totally different from the other algorithms, is it a typo?
  13. Could the authors make the results and Matlab code available for reviewers to check the correctness of results? Also what are the specific implementations of the FA and PSO algorithms?

Author Response

The authors are grateful to the reviewer for the insightful and constructive comments and suggestions that have enhanced the revised manuscript. The extent of detail in the reviewer’s comments is quite impressive. We have carefully addressed all comments as explained in the attached response to comments, and we revised the manuscript accordingly. All changes made to address the reviewer’s comments are highlighted in the revised manuscript and can easily be identified.

Reviewer 2 Report

-Check grammar of the long adjective and rewrite the phrase:  "In the past, retaining structures' initial anticipated dimensions were tested for stability and other building code requirements."
-Insert a citation in the places where you claim that Salp Swarm Optimization outperform other algorithms.
-When presenting Objective function in Equation 8, please indicate whether the parameters are non-linear,  I mean, in the current form it seems that the objective function is completely linear, hence, there is no motivation to use a - metaheuristic.

-Indicate in the algorithm how the design constraints are managed.

-In table 1, use the literals in Equation 8 to indicate what, of the presented cost, each one is.

- I think that it is better if you present all the design problems in a single section, instead of 2. And to unify tables 3 and 6, because some literals are used in both sections currently.
- I think that the results with benchmark functions are promising, nevertheless, I think they are not fair, because the comparing algorithms do not report the best performance for 30 individuals and 1000 iteration, the population size is to small for most of them and the number of iterations is a lot, hence I think that this parameters favors the proposal. It is not completely incorrect to show that your proposal requires less memory than others, but you must not claim that it is a "fair" comparison. Present hypothesis test in the comparison.

- For the design problem you need to perform several executions of the algorithm and to present hypothesis test.

In general, I think that you must rearrange the presentation, presenting first the comparison with benchmark functions, then the results with the structural design problem, and trying to increase the clarity of the design problem, including the justification/motivation of using a metaheuristic on this problem.

The conclusion must be improved. You actually reminded the paper in a summarized way, but you did not present a conclusion, for instance, what is the cause behind outperforming the comparing algorithm? If it is the diversity management or a exploration technique, can you provide some evidence? For instance, you can provide a variance graph or a distance measure among individuals. 

Author Response

The authors would like to thank the reviewer for the insightful and constructive comments and suggestions that have enhanced the revised manuscript. The extent of detail in the reviewer’s comments is quite impressive. We have carefully addressed all comments as explained in the attached response to comments, and we revised the manuscript accordingly. All changes made to address the reviewer’s comments are highlighted in the revised manuscript and can easily be identified.

Reviewer 3 Report

The manuscript is well written, I just suggest standardizing the number of decimal places in the tables.

I recommend publication in Applied Sciences.

Author Response

Thank you for this valuable comment. The authors considered the reviewer's recommendation in the revised paper and the number of decimal places in Tables 4 and 5 were modified.

We would like to appreciate your time and valuable review of our work. We hope that the revised manuscript is satisfactory and meets your expectations.

Reviewer 4 Report

You need to mention your previous papers on the topic, and explicitly show how the work presented in this manuscript is different and novel. This is because I've seen that you do publish a lot: just for 2022 Google reports 14 art. MK, 20 art. MLN, 39 art. SK, etc. A quick inspection reveals plenty of similar articles published just in 2022, appearing in IEEE Access, Sustainability, Structural Eng. Mech., Structures, Eng. Struct., Intl. J. Impact Eng., etc. None of these is mentioned in the list of references, the only 2 references to your previous work being [4] from 2011, and [14] from 2021.

I've checked only one publication from 2022 (A Arabali, M Khajehzadeh, S Keawsawasvong, AH Mohammed, B Khan, "An Adaptive Tunicate Swarm Algorithm for Optimization of Shallow Foundation," IEEE Access, 4 Apr. 2022; 10.1109/ACCESS.2022.3164734) and differences should have been detailed as figures and tables are similar, with some practically identical. This raises the question of copyrights which have already been given to IEEE, but I have not seen any mentioning (either in captions or in footnotes) that permission to reproduce was granted by IEEE. True, if publishing a few tens of papers per year, timing makes it that you are not sure when submitting the manuscript if the other manuscripts are going to be accepted. Still, in this particular case it looks like the manuscript was submitted in April 23, while the IEEE Access article was online on April 4. 

Author Response

The authors are grateful to the reviewer for the insightful and constructive comments and suggestions that have enhanced the revised manuscript. The extent of detail in the reviewer’s comments is quite impressive. We have carefully addressed all comments as explained in detail in the attached file and revised the manuscript accordingly. All changes made to address the reviewer’s comments are highlighted in the revised manuscript and can easily be identified.

Round 2

Reviewer 2 Report

I think that the authors "must" perform hypothesis test and report the p-value over executions of "all" problems, this is the same suggestion than in my previous review. Hypothesis tests are not time consuming if you have the execution's data and provide statistical significance to the claims about superiority.  

Author Response

We would like to thank the reviewer for the objective and thorough review of our paper. We have carefully addressed the second round of  comments and revised the manuscript accordingly. All changes made to accommodate the reviewers’ comments are highlighted in the revised manuscript.

As recommended, the authors performed hypothesis testing . Accordingly, the nonparametric Wilcoxon's rank sum test was performed for benchmark functions on pages 12 –13 and for design examples on pages 22 – 23.

We would like, once again, to appreciate your time and thoughtful review of our work. We hope that that this second revision meets publication standards..

Round 3

Reviewer 2 Report

All suggestion where considered. I recommend to accept the article in its current form.